# Signal-to-Noise Analysis Can Inform the Likelihood That Incidentally Identified Variants in Sarcomeric Genes Are Associated with Pediatric Cardiomyopathy

**DOI:** 10.3390/jpm12050733

**Published:** 2022-04-30

**Authors:** Leonie M. Kurzlechner, Edward G. Jones, Amy M. Berkman, Hanna J. Tadros, Jill A. Rosenfeld, Yaping Yang, Hari Tunuguntla, Hugh D. Allen, Jeffrey J. Kim, Andrew P. Landstrom

**Affiliations:** 1Division of Pediatric Cardiology, Department of Pediatrics, Duke University School of Medicine, Durham, NC 27710, USA; leonie.kurzlechner@duke.edu (L.M.K.); amy.berkman@duke.edu (A.M.B.); 2Section of Pediatric Cardiology, Department of Pediatrics, Baylor College of Medicine, Houston, TX 77030, USA; edward.jones@bcm.edu (E.G.J.); hanna.tadros@bcm.edu (H.J.T.); hari.tunuguntla@bcm.edu (H.T.); hdallen@texaschildrens.org (H.D.A.); jjkim@texaschildrens.org (J.J.K.); 3Department of Pediatrics, University of Florida, Gainesville, FL 32611, USA; 4Department of Molecular and Human Genetics, Baylor Genetics Laboratories, Baylor College of Medicine, Houston, TX 77030, USA; jill.mokry@bcm.edu (J.A.R.); yapingy@bcm.edu (Y.Y.); 5Department of Cell Biology, Duke University School of Medicine, Durham, NC 27710, USA

**Keywords:** hypertrophic cardiomyopathy, exome sequencing, incidentally identified variant, secondary finding, genetic testing, mutation hotspot

## Abstract

Background: Hypertrophic cardiomyopathy (HCM) is the most common heritable cardiomyopathy and can predispose individuals to sudden death. Most pediatric HCM patients host a known pathogenic variant in a sarcomeric gene. With the increase in exome sequencing (ES) in clinical settings, incidental variants in HCM-associated genes are being identified more frequently. Diagnostic interpretation of incidental variants is crucial to enhance clinical patient management. We sought to use amino acid-level signal-to-noise (S:N) analysis to establish pathogenic hotspots in sarcomeric HCM-associated genes as well as to refine the 2015 American College of Medical Genetics (ACMG) criteria to predict incidental variant pathogenicity. Methods and Results: Incidental variants in HCM genes (*MYBPC3*, *MYH7*, *MYL2*, *MYL3*, *ACTC1*, *TPM1*, *TNNT2*, *TNNI3*, and *TNNC1*) were obtained from a clinical ES referral database (Baylor Genetics) and compared to rare population variants (gnomAD) and variants from HCM literature cohort studies. A subset of the ES cohort was clinically evaluated at Texas Children’s Hospital. We compared the frequency of ES and HCM variants at specific amino acid locations in coding regions to rare variants (MAF < 0.0001) in gnomAD. S:N ratios were calculated at the gene- and amino acid-level to identify pathogenic hotspots. ES cohort variants were re-classified using ACMG criteria with S:N analysis as a correlate for PM1 criteria, which reduced the burden of variants of uncertain significance. In the clinical validation cohort, the majority of probands with cardiomyopathy or family history hosted likely pathogenic or pathogenic variants. Conclusions: Incidental variants in HCM-associated genes were common among clinical ES referrals, although the majority were not disease-associated. Leveraging amino acid-level S:N as a clinical tool may improve the diagnostic discriminatory ability of ACMG criteria by identifying pathogenic hotspots.

## 1. Introduction

Hypertrophic cardiomyopathy (HCM) is the most common heritable cardiomyopathy, predominantly exhibiting autosomal dominant inheritance. The prevalence of HCM in the general population is estimated to be 1:500, and many HCM patients have a known pathogenic variant [1,2,3]. HCM is a heterogeneous disease, ranging from no symptoms to sudden cardiac death. In fact, HCM is the prevailing cause of sudden death in young people, including athletes. Other manifestations include diastolic dysfunction, tachyarrhythmias, thromboembolic events, and progressive heart failure. Annual mortality risk for all patients with HCM ranges from 1.5 to 6.0% [1,4,5]. In pediatric populations, HCM-related mortality has a bimodal peak in the first two years of life and in adolescence, although many patients have little disability and normal life expectancy [1,4,5]. Clinical diagnosis of HCM is made by echocardiogram, with identification of unexplained left ventricular wall thickening together with a non-dilated cavity [6,7].

The American Heart Association and American College of Cardiology recommend genetic testing for individuals with a likely diagnosis of HCM and first-degree relatives of HCM probands found to be genotype-positive to guide screening and monitoring of affected individuals and family members [8,9,10]. Sarcomeric genes are the major genetic cause of HCM and encode proteins making up basic contractile units of the cardiac myocyte. These include *MYH7*-encoded beta-myosin heavy chain, *MYBPC3*-encoded cardiac myosin-binding protein C, *TNNT2*-encoded cardiac troponin T, *TNNI3*-encoded cardiac troponin I, *ACTC1*-encoded cardiac actin, *TPM1*-encoded alpha-tropomyosin, *MYL2*-encoded regulatory myosin light chain, *MYL3*-encoded essential myosin light chain, and *TNNC1*-encoded cardiac troponin C [11]. These genes constitute the major portion of diagnostic gene panel testing for patients with suspicion of HCM.

In contrast to diagnostic gene panel testing, exome sequencing (ES) detects variants in every coding area of the genome. Due to its ability to diagnose individuals with atypical presentations of Mendelian disorders, the clinical use of ES is increasing [12]. While there is a clear diagnostic role for ES, incidental genetic findings, including those localizing to cardiovascular disease-associated genes, can complicate ES result interpretation. Based on 2015 ACMG criteria, these variants can be assigned a likely pathogenic (LP) or pathogenic (P) disease association, or, if association with disease is unclear, labeled as a variant of uncertain significance (VUS) [13]. Incidental findings present a clinical dilemma when found in genes associated with life-threatening conditions, especially when there is no pre-test suspicion for disease. Community-based cohorts and ostensibly healthy individuals exhibit a rare variant frequency of 5–11% in HCM-associated genes [14,15], which is markedly higher than the disease frequency in the population (1 in 500) and highlights the challenges in interpreting these variants [13,16]. Given the morbidity and mortality of HCM and the broad utilization of the 2015 ACMG criteria to assign pathogenicity to incidental variants in HCM-associated genes, we sought to refine this analysis for improved predictive potential—in particular for incidentally identified VUSs. 

This study uses amino acid-level signal-to-noise (S:N) analysis, a method of identifying pathogenic hotspots within a gene, to establish disease-associated “hotspots”. We used this analysis to predict pathogenicity of incidental variants in HCM genes found on clinical ES testing and applied the results of this analysis to augment ACMG variant classification. Furthermore, we describe the clinical histories and echocardiographic findings of carriers of incidental variants to validate a refined model of ACMG variant classification in patients with pediatric-onset cardiomyopathy.

## 2. Methods

Study Cohorts

This study was approved by the Duke University School of Medicine and Baylor College of Medicine institutional review boards. All participants provided informed consent in accordance with the Declaration of Helsinki. All other methods are detailed in the Appendix A. 

## 3. Results

### 3.1. Rare Variant Prevalence in HCM-Associated Genes among ES Referrals

Incidentally identified likely pathogenic/pathogenic (LP/P) variants and VUSs localizing to sarcomeric HCM genes were identified in ES referrals to establish a “background” incidental variant rate. There were 7244 individuals in the ES cohort, including 3909 (54%) males, 3274 (45.2%) females, and 61 (0.8%) fetal subjects (Table 1). Following exclusion of fetal samples and kindred samples, 7066 unrelated probands were identified with a median age at referral for genetic testing of 6.1 years [2.5–12.3] (Figure 1A). Among ES probands, 509 (7.2%) individuals hosted at least one rare variant in an HCM-associated sarcomeric gene yet concern for cardiovascular disease was the indication for genetic testing in a minority (2.6%) of variant-positive probands. There were six individuals referred solely for an indication of hypertrophic cardiomyopathy for whom identification of a LP/P variant may be diagnostic. These individuals comprised ~1% of the total number of index cases hosting incidentally identified genetic variants. Most variant-positive probands hosted a variant ascribed to be a VUS (6.7% of ES cohort) at time of genetic test reporting, while 0.5% hosted an LP/P variant (Figure 1B). Most probands hosted a single variant (93.7%), while a subset (6.3%) hosted two variants including both LP/P and/or VUS (Figure 1C). Of the 509 ES probands, trio data was available for 64, of which 6 (9.4%) hosted a confirmed de novo variant (Appendix A). Overall, these findings suggest that 0.5% of ES referrals will have an incidentally identified LP/P variant in a sarcomeric HCM-associated gene while a significant ~7% will have an incidentally identified VUS despite a low pre-test clinical suspicion of cardiomyopathy.

### 3.2. ES Variant Frequency Compared to Control and HCM-Afflicted Individuals

Given the high rate of incidentally identified variants in the ES cohort, we compared these findings to the prevalence of rare variants found in HCM-associated genes among control (ostensibly healthy) subjects from the gnomAD cohort and a disease cohort of individuals with HCM. Among 138,632 individuals in the gnomAD cohort, there were 7247 variants with a minor allele frequency (MAF) <0.0001 localizing to an HCM-associated sarcomeric gene of interest, yielding a gnomAD rare variant frequency of 5.3% (Figure 2A). In the literature-derived HCM case cohort, at least one sarcomeric gene variant was identified in 2877 individuals with HCM (41.1%, Appendix A). The yield of sarcomeric gene-positive individuals in the HCM cohort was 5.7-fold higher than in the ES cohort (*p* < 0.0001). Conversely, the distribution of variant frequencies by gene in the ES cohort was similar to the gnomAD control (Sign test, *p* = 0.18) and statistically dissimilar to the HCM cohort (*p* < 0.01). There was also a statistically significant difference in the distribution of variant frequencies by gene between the control and HCM cohorts (*p* < 0.05, Appendix A, Figure 2B). We next stratified our analysis by variant type (missense versus radical). The ES and gnomAD cohorts exhibited a predominance of missense variants. The HCM cohort demonstrated a nearly 20-fold higher frequency of radical variants compared with ES (*p* < 0.05) and control (*p* < 0.05) cohorts. Our evaluation of gene-specific variant prevalence stratified by variant type and cohort can be found in the Appendix A (Appendix A). These findings indicate that variants in sarcomeric genes are markedly enriched in the HCM cohort when compared to both the gnomAD and ES cohorts. Overall, missense variants are the most common in all cohorts, but radical variants have a 20-fold higher frequency in the HCM cohort relative to ES referrals and gnomAD controls. Furthermore, gene-specific variant frequencies are similar between ES and gnomAD controls.

### 3.3. Gene-Level Signal-to-Noise in ES and Pathogenic HCM Cohort

We have previously shown that the relative frequency of pathogenic versus population variants can identify genetic “hotspots” where disease-associated variants are probabilistically more likely to be located [17,18]. Given the relatively high prevalence of background genetic variation in HCM-associated genes among control individuals, we calculated gene-specific signal-to-noise relative minor allele frequency (S:N ratio) by normalizing the minor allele frequency of variants in ES and HCM cohorts, respectively, against the population minor allele frequency in gnomAD. Five out of nine disease-associated genes carried an HCM:gnomAD ratio greater than 5.0, with the highest S:N in *MYBPC3* (12.6) and *MYH7* (9.2, Figure 3A), suggesting a higher relative frequency of disease-associated variants in these genes, overall. Conversely, among ES referrals, no gene had a S:N greater than 5.0, with more modest S:N values with the highest being *MYL2* (2.3), *MYBPC3* (2.0), and *TNNT2* (2.0, Figure 3B). Overall, we found a significantly higher S:N ratio in *MYBPC3*, *MYH7*, *TNNI3*, *TNNT2*, and *MYL2* in the HCM cohort when compared to ES referrals. Overall, this suggests that the five genes with a S:N greater than 5.0 have the highest probability of pathogenic disease-associated variants versus physiologically tolerated, population variants. 

### 3.4. Amino Acid-Level Signal-to-Noise to Inform ACMG Pathogenicity Criteria

Given the large disparity between S:N among disease-associated variants and ES-associated variants, we next sought to apply S:N analyses at the amino acid-level to establish a high-resolution map of disease-associated variant “hotspots”. Moreover, we have recently shown that S:N-identified hotspots can be used to augment the 2015 ACMG criteria [19] for classifying variants diagnostically. To do this, we calculated a gene-specific S:N threshold of pathogenicity above which amino acid positions were considered more likely to host pathogenic variants. This threshold was set to 2.40 for *MYBPC3*, 2.58 (*MYH7*), 2.19 (*MYL2*), 2.02 (*MYL3*), 1.70 (*ACTC1*), 2.23 (*TPM1*), 3.86 (*TNNT2*), 4.93 (*TNNI3*), and 1.37 (*TNNC1*). Amino acid positions above threshold are listed in Appendix A. The largest HCM:gnomAD peaks which rose above this threshold were found in *MYH7*, *TPM1*, *TNNI3*, and *MYBPC3*. Thin filament genes (*TNNT2*, *ACTC1*, and *TNNC1*) had far fewer peaks, with the exception of *TNNT2*, which had S:N peaks that were both wider and taller than those in *ACTC1* or *TNNC1.* A detailed stratification of S:N analysis by variant type and domain-specific S:N analyses of minor genes that are not common genetic causes of HCM are further detailed in the Appendix A (Appendix A). 

In *MYH7*, nearly all pathogenic amino acid positions were in the head domain, between amino acids 169 and 982 (Figure 4A, Appendix A). Two areas of high signal overlapped with actin binding sites at amino acids 657–671 and 762–768. Given that these sites are crucial for sarcomere function and have high S:N in the disease cohort [20,21], variants at these sites may have a high probability of being pathogenic. However, in *MYBPC3*, pathogenic S:N regions did not track to any domain type (Figure 4B, Appendix A). In contrast to HCM-associated variants, when normalizing the ES cohort variants in a similar manner, there were comparatively lower amino acid-level S:N ratios for variants in both *MYH7* and *MYBPC3*. Low S:N ratios in the ES cohort indicate that variants at any given location in *MYH7* and *MYBPC3* are not found at a higher prevalence in individuals undergoing ES relative to a large population cohort, gnomAD. Taken together, these findings indicate that topographical distribution of HCM-case variation is dependent both on disease state (HCM versus control) and varied by the gene. Furthermore, amino acid location of variants is more likely to be predictive of pathogenicity in *MYH7* than in *MYBPC3*. 

### 3.5. Incorporation of Signal-to-Noise Analysis into Variant Interpretation

We next applied these S:N thresholds to the ACMG PM1 criteria for variant re-evaluation. Specifically, we considered amino acid positions with HCM:gnomAD S:N exceeding the gene-specific threshold as meeting PM1 criteria. Incidentally identified VUSs were then re-evaluated based on revised ACMG criteria, incorporating PM1 criteria when applicable. On re-evaluation of these VUSs, we found that most variants shifted in classification to LP/P (Figure 5A). Initially, 96% of variants (N = 520) were classified as a VUS and 4% as LP/P without the PM1 criteria applied (N = 21). Following reclassification, the proportion of variants interpreted as a VUS dropped to 84% (N = 455, *p* < 0.001), with 15.9% now classified as LP/P (N = 86). Appendix A includes the full list of ES variants and their classification by Baylor Genetics Laboratories (at time of genetic testing), our initial classification with ACMG criteria, and our re-evaluation incorporating S:N analysis. These findings suggest that leveraging amino acid-level S:N may improve the diagnostic discriminatory ability of the ACMG criteria.

### 3.6. Clinical Validation of Incidentally Identified Variants Re-Assigned as Likely Pathogenic 

To determine whether pathogenicity re-assignment was diagnostically valid, we compared the presence of pediatric-onset cardiomyopathy in individuals with incidentally identified LP/P versus VUS variants. To do this, we established a clinical validation cohort of individuals with ES variants that underwent cardiac evaluation with echocardiogram. We retrospectively reviewed individuals in the ES cohort who were clinically evaluated at Texas Children’s Hospital (TCH). The vast majority of these probands were not referred solely for cardiovascular issues (97.4%, Appendix A). Of 509 variant-positive probands in the ES cohort, 171 (33.6%) were clinically evaluated at TCH, with 148 probands after exclusion criteria and 38 (7.5%) subjects undergoing cardiac and echocardiographic evaluation (Appendix A, Figure 5B). As expected, the age at genetic testing for the TCH cohort was young, with an average age at genetic testing of 6.8 years. Of the nine genes analyzed, only variants in *MYBPC3*, *MYH7*, *TPM1*, *TNNT2*, *MYL2*, *MYL3*, and *TNNI3* were seen in the cohort clinically evaluated at TCH. Among the 28 individuals in the TCH cohort hosting a LP/P ES variant (after reclassification), 6 (21%) had pediatric-onset cardiomyopathy diagnoses (excluding mitochondrial cardiomyopathies). Of these six patients with cardiomyopathy, two had a variant initially classified as a VUS prior to applying S:N criteria. 

Notably, following re-interpretation, of 120 patients hosting a VUS, only 2 had clinical evidence of pediatric cardiomyopathy (1.7%) compared to 6/28 (21%) patients hosting LP/P variants (*p* < 0.001, Figure 5C). Following reclassification with S:N, six patients with cardiomyopathy or family history hosted an LP/P variant (Table 2) and the remaining two cardiomyopathy cases hosted VUSs. Of the eight individuals with cardiomyopathy or a family history, 75.0% hosted an LP/P variant, while 15.7% of individuals with a negative clinical evaluation hosted an LP/P variant (*p* < 0.001, Figure 5D). Some individuals hosting an LP/P variant may be phenotype negative at time of evaluation but develop disease later in life, which is not reflected in our study. Additional analysis of pre-ES clinical suspicion of cardiomyopathy, variant frequency, and clinical description of probands with evidence of cardiomyopathy are further detailed in the Appendix A. Taken together, these findings suggest that incorporation of S:N analysis in ACMG criteria can be diagnostically informative.

## 4. Discussion

We demonstrate that amino acid-level S:N analysis can aid in predicting pathogenicity of incidental variants in sarcomeric HCM-associated genes. Clinically, ES is a valuable tool given its sensitivity and ability to offer diagnostic guidance when individuals exhibit a nonspecific phenotype or heterogeneous disease presentation [22]. The expanding availability and use of ES clinically and direct-to-consumer genetic testing often prompts an evaluation of how to interpret incidentally identified variants in genes deemed clinically actionable by ACMG guidelines [13,23]. With increased genome-wide testing, more variants in genes associated with cardiac channelopathies and cardiomyopathies are incidentally identified. This is not an uncommon problem, as previous work has demonstrated a 34-fold increased prevalence of incidental VUSs compared to HCM disease prevalence [22]. 

ACMG criteria may be used to evaluate the pathogenicity of incidental variants, which can aid in facilitating patient management and cascade genetic testing of family members. Our work suggests that incorporation of S:N analysis may improve the diagnostic accuracy of the existing ACMG criteria. Moreover, this analysis has previously been demonstrated to be useful in pathogenicity interpretation of incidental VUSs in *TTN* truncating variants and variants associated with cardiac channelopathies [17,18,24,25]. While current ACMG guidelines recommend against the clinical reporting of incidentally identified VUSs as medically actionable due to their low positive predictive value, there are a number of mechanisms by which VUSs may yet be reported, including variant classification changing over time or inappropriate gene panel testing [23,26]. Developing tools to better predict VUS pathogenicity is essential given the medical management and psychosocial consequences associated with inherited cardiomyopathies. This is especially relevant in HCM, as we found an incidentally identified HCM-associated rare variant in 5% of controls and variants classified as VUS in 7% of ES referrals, suggesting a high degree of healthy genetic variation in these genes. 

As demonstrated by amino acid-level S:N analysis, variant location may have pathogenicity implications. For example, in *MYH7*, one of two sarcomeric genes most commonly associated with HCM, previous studies found that disease-associated variants cluster in the myosin head domain, especially the motor region, which is essential for cardiomyocyte structure and movement [27,28]. We similarly found that the myosin head domain and actin binding regions are LP/P variant hotspots, while variants in other regions likely represent healthy genetic variation. In addition, modified criteria for *MYH7* established by the Clinical Genome Resource (ClinGen) have helped refine the clinical utility of current ACMG guidelines [29,30]. The validity of these revisions in dilated cardiomyopathy underscores the significance of similar changes to the interpretation of variants in HCM-associated genes [31]. 

While we did not find domain-specific clustering of pathogenic hotspots in *MYBPC3*, we did find differences by variant type. To account for the possibility of errors of interpretation that could arise from applying an elevated S:N caused by pathogenic radical variants (predicted to alter more than a single amino acid and may cause early termination, insertion/deletion, and splice site changes) to an identified missense variant, or vice versa, we elected to stratify S:N hotspot analysis by variant type. Radical variants in *MYBPC3* were more common in the HCM than the ES or gnomAD cohorts and were predominantly found to have S:N ratios exceeding the significance threshold. S:N analysis limited to only missense variants may help avoid any bias introduced by radical variants, which may abolish protein structure. Similarly, previous studies have found that most LP/P variants in this gene are radical variants [32]. In *MHY7*, we did not see an outsized impact of radical variants on pathogenicity prediction. 

In addition to using amino acid-level S:N analysis across known disease and control cohorts, we used this analysis with ACMG criteria to reclassify incidental variants in ES referrals who underwent clinical evaluation at TCH. As there is overlap in the types of cardiomyopathies that are associated with variants in sarcomeric genes, we identified not only HCM, but also dilated cardiomyopathy and left ventricular non-compaction cardiomyopathy as cases in this cohort [33,34]. Before reclassification of variants, a disease-causative (LP/P) variant was not found in half of cardiomyopathy cases. While cardiomyopathy was rare in this cohort, most cardiomyopathy cases hosted a variant classified as LP/P after ACMG re-interpretation with PM1 applied. Notably, some individuals hosting an LP/P after re-analysis may be phenotype negative at time of genetic testing but may develop disease over time (particularly relevant given the young age of the cohort). Therefore, our retrospective chart review does not catch all cardiomyopathy cases. This further supports the utility of S:N analysis in refining pathogenicity of incidentally identified VUS in sarcomeric HCM-associated genes. 

Given that we did see an overall increase in LP/P variants after VUS reassignment with PM1 criteria, additional studies are needed to define the diagnostic weight of all incidentally identified sarcomeric variants. The methods used for our re-classification only add ACMG criteria, therefore, variants could only be upgraded in classification. S:N analysis may also be incorporated in future studies by incorporating a lower threshold for S:N, below which variants may instead be assigned criteria for reclassifying VUS to benign/likely benign. Should additional work continue to support the diagnostic strength of S:N analysis, it may be leveraged as a clinical tool to prospectively determine a child’s risk of developing HCM and thus inform screening recommendations when VUS are incidentally found. 

Given the young age of the cohort that underwent clinical evaluation, additional individuals may develop cardiomyopathy over time. Thus, our analysis of the clinical validation cohort at TCH would not be able to account for individuals with later onset cardiomyopathy as the study only evaluated individuals at one time point. Moreover, at the time that sequencing was performed for most ES referrals, Baylor Genetics Laboratories did not emphasize trio analyses, which limits conclusions which can be drawn about the mode of inheritance of the identified variants. We were also unable to identify gnomAD individuals hosting multiple variants, leading to a potential over count for some cohort allele comparisons. However, a strength of the study is the large population of ES referrals and gnomAD controls, allowing for comparison of these cohorts with a known disease cohort and reducing the impact of gnomAD individuals hosting several variants. 

Future work should focus on longitudinal studies to evaluate whether individuals incidentally identified as hosting an LP/P sarcomeric gene variant developed cardiomyopathy later in life. This would be done most rigorously as a prospective study with robust clinical follow-up. Understanding the correlation between variant pathogenicity or type and age of disease onset would allow for assessment of penetrance and expressivity. Future studies would benefit from a closer look at the inheritance of incidentally identified variants. Assessing phenotype-genotype correlations in families would also offer valuable insights on disease penetrance and expressivity. 

In conclusion, incidental variants in sarcomeric HCM-associated genes in the general population and ES referrals are relatively frequent and the majority are not disease-associated, thus likely reflect background genetic noise in the absence of pre-test probability. Based on amino acid-level S:N analysis, incidentally found variants in the clinical ES testing cohort were more similar to incidental variants found in the general population, while variants identified in known HCM cases had different localization patterns. Most incidental variants in HCM-associated sarcomeric genes in the clinical ES cohort were not associated with signs of cardiomyopathy, and among those that were, there was high pre-test probability of disease. For the small number of VUSs that are disease-associated, leveraging S:N analysis to enhance ACMG pathogenicity criteria may aid in variant classification, and, ultimately, clinical management. Given the high prevalence of rare variants in HCM-associated genes among the healthy population, accurate interpretation of genetic testing necessitates frequent revision, which can be improved by tools such as gene- and amino acid-level S:N ratios that leverage population-based genetic studies. Using these studies increases the diagnostic utility of next-generation sequencing modalities, decreases the chance of misdiagnosis, and informs the need for clinical follow-up.

## Figures and Tables

**Figure 1 jpm-12-00733-f001:**
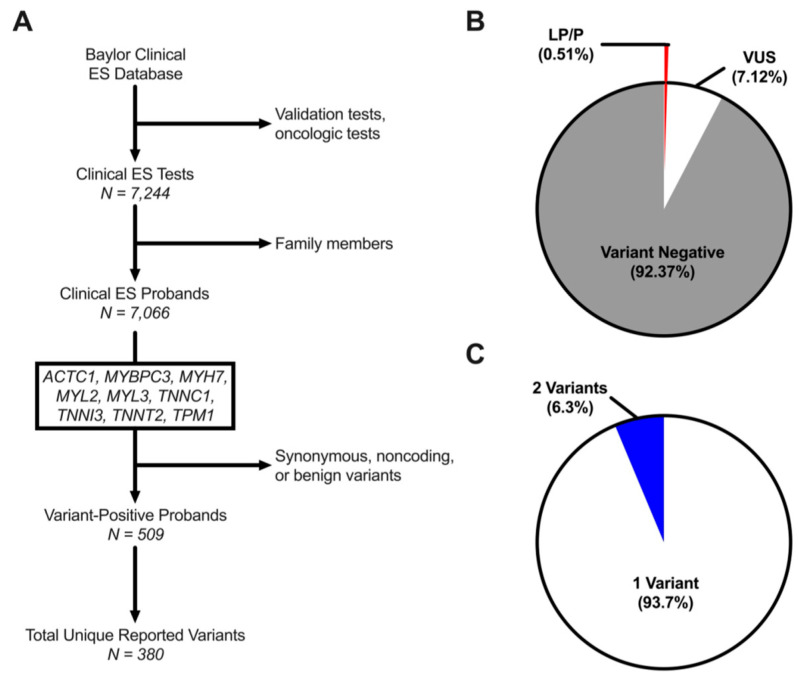
(**A**), Schematic of the study methodology. (**B**), Pie chart of all subjects undergoing ES testing divided into individuals with no HCM gene-associated variant (variant-negative; grey) and individuals who hosted a variant of uncertain significance (VUS; white) or likely pathogenic/pathogenic variants (LP/P, red) according to laboratory interpretation at the time of reporting. (**C**), Pie chart of ES cohort demonstrating variant-positive individuals with a single variant (white) and those with two or more variants (blue).

**Figure 2 jpm-12-00733-f002:**
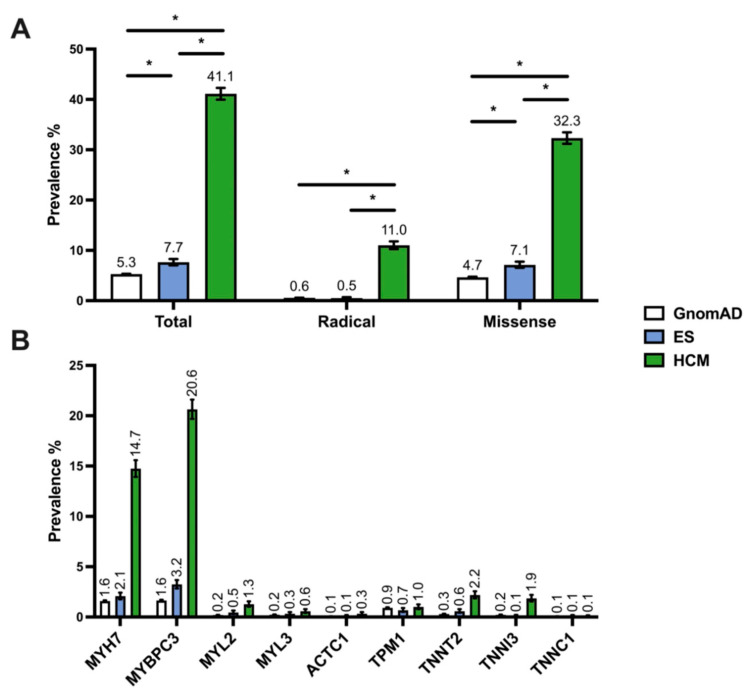
(**A**), Bar graph comparing total, radical, and missense variant frequency of HCM-associated variants. (**B**), Bar graph comparing variant frequency across genes. Comparisons made across a control cohort (gnomAD, white), a cohort of ES referrals (blue), and a cohort of individuals with clinical HCM (green). Error bars denote 95% CI. *, *p* < 0.05.

**Figure 3 jpm-12-00733-f003:**
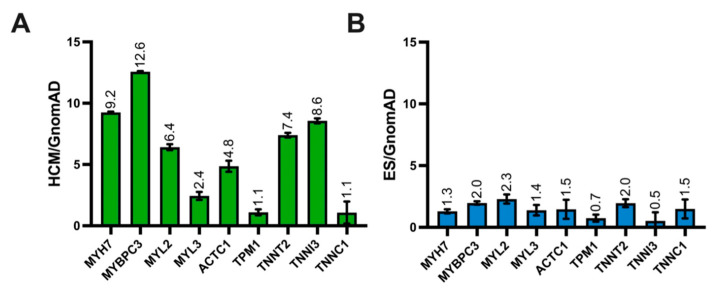
(**A**), Bar graph demonstrating the gene-level signal-to-noise ratios for each HCM-associated gene for the HCM case cohort compared with the gnomAD cohort. (**B**), Bar graph demonstrating the signal-to-noise ratios for each HCM-associated gene for the ES cohort compared with the gnomAD cohort. Error bars denote 95% CI.

**Figure 4 jpm-12-00733-f004:**
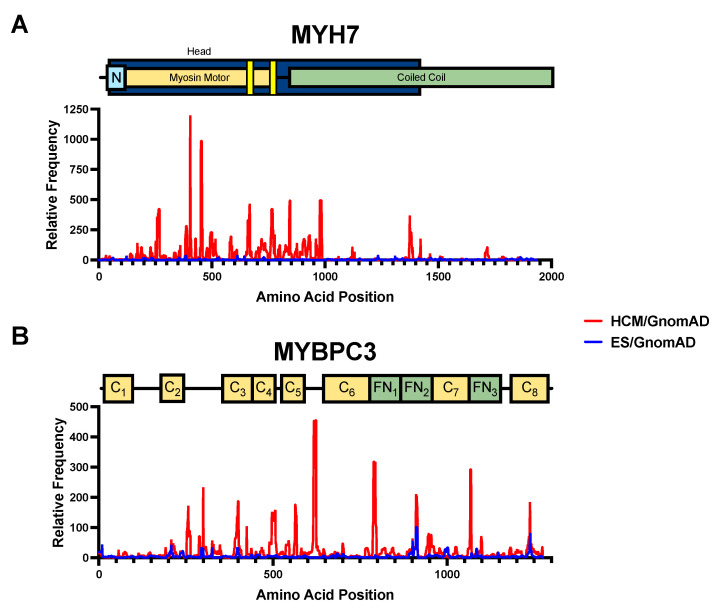
(**A**), Amino acid-level signal-to-noise analysis of *MYH7* variants for both HCM cases (red) and ES-identified variants (blue), compared with variants identified in the gnomAD cohort. (**B**), Amino acid-level signal-to-noise analysis of *MYBPC3* variants for both HCM cases (red) and ES-identified variants (blue), compared with variants identified in the gnomAD cohort. Functional domains of *MYH7* and *MYBPC3* are depicted. N, N-terminal domain; Cx, immunoglobulin-like domain; FNx, fibronectin domain; highlighted yellow regions, actin binding sites.

**Figure 5 jpm-12-00733-f005:**
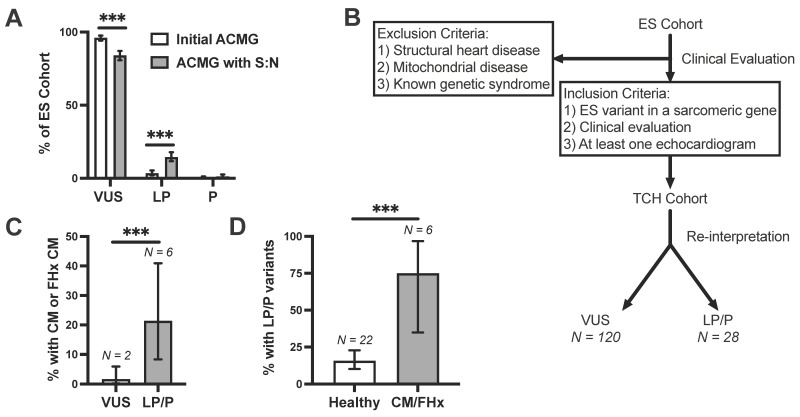
(**A**), Bar graph demonstrating proportion of pathogenic (P), likely pathogenic (LP), and variants of uncertain significance (VUS) using American College of Medical Genetics and Genomics (ACMG) criteria before and after re-assignment with signal-to-noise (S:N). (**B**), Exome sequencing (ES) cohort was assigned pathogenicity based on 2015 ACMG criteria. A retrospective clinical analysis was performed on those patients seen at Texas Children’s Hospital (TCH) following exclusion of subjects with structural heart disease, mitochondrial disease, or chromosomal abnormalities, or those without echocardiogram. (**C**), Bar graph showing the percentage of individuals with cardiomyopathy (CM) or first-degree family history (FHx) of cardiomyopathy out of TCH cohort probands hosting VUSs or LP/P variants. (**D**), Bar graph highlighting the percentage of individuals hosting LP/P variants out of TCH cohort probands with a negative clinical evaluation or cardiomyopathy or first-degree family history of cardiomyopathy. ***, *p* < 0.001.

**Table 1 jpm-12-00733-t001:** Exome sequencing cohort demographics at time of genetic testing.

Characteristic	N (%)
Total Individuals	7244
Male	3909 (54.0%)
Female	3274 (45.2%)
Fetal	61 (0.8%)
Age at Genetic Testing (y)	6.1 [2.5–12.3]
Total Probands	7066
Unique Variants	380
LP/P	26 (6.8% [4.3–9.4])
VUS	354 (93.2% [90.6–95.7])
Variant-Positive Probands	509 (7.2% [6.6–7.8])
1 Variant	477 (93.7% [91.6–95.8])
2 Variants	32 (6.3% [4.2–8.4])

y, years of age; LP/P, likely pathogenic/pathogenic; VUS, variant of uncertain significance.

**Table 2 jpm-12-00733-t002:** TCH cohort variants.

Probands Hosting One Variant											
Gene	Sex	Age (y)	Ethnicity	Initial Classification	Reclassification with S:N	Nucleotide	Amino Acid	Variant Type	Zygosity	Cardiac Problem	Family Cardiac Hx
*MYH7*	F	9.01	African	VUS	VUS	3301G > A	G1101S	Missense	Het	No	CM death in father at 32
*MYBPC3*	F	0.06	Caucasian	VUS	LP	2063C > T	T688M	Missense	Het	Unspecified CM	No
*MYBPC3*	F	15.8	Hispanic	VUS	VUS	3415G > A	V1139I	Missense	Het	CM, muscular dystrophy	No
*TPM1*	F	1.59	African	P	P	475G > A	D159N	Missense	Het	LVNC	No
*TPM1*	M	0.54	African	P	P	688G > A	D230N	Missense	Het	LVNC	No
*TNNT2*	F	0.57	Hispanic	LP	LP	391C > T	R131W	Missense	Het	DCM	No
*TNNI3*	F	7.68	Caucasian	VUS	LP	400G > C	D134H	Missense	Het	DCM	No
**Proband Hosting Two Variants**											
*MYBPC3*	M	6.05	Hispanic	VUS	VUS	2915G > A	R972Q	Missense	Het	DCM	Mother with DCM
*MYH7*	M			LP	LP	2710C > T	R904C	Missense	Het		

CM, cardiomyopathy; DCM, dilated cardiomyopathy; F, female; HCM, hypertrophic cardiomyopathy; Het, heterozygous; Hx, history; LP, likely pathogenic; LVNC, left ventricular noncompaction; M, male; P, pathogenic; VUS, variant of uncertain significance; y, years.

## Data Availability

The data represented in this study are contained within this manuscript, the Appendix A, or available on reasonable request from the corresponding author. Data is not publicly available due to use agreement with Baylor Genetics Laboratories.

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
