# Peer review of "Signal-to-Noise Analysis Can Inform the Likelihood That Incidentally Identified Variants in Sarcomeric Genes Are Associated with Pediatric Cardiomyopathy"

_jpm, 2022, doi:10.3390/jpm12050733_

Round 1

Reviewer 1 Report

In this manuscript, Kurzlechner and colleagues compared the amino acid-level signal-to-noise (S:N) in nine HCM-associated sarcomeres genes (MYBPC3, MYH7, MYL2, MYL3, ACTC1, TPM1, TNNT2, TNNI3, TNNC1) in three subjects cohorts, a 7,244 ES, 2,912 HCM and the gnomAD controls. The study is very important to determine the clinical burden of genetic variability in those genes and distinguish between genetic constraint and tolerated changes. According to the authors, “incidental variants in HCM-associated genes were common among clinical ES referrals, though the majority were not disease-associated.” 

However, there are few points to be addressed:

In Supplementary Table 4, the authors listed the variants identified in the ES cohort and their classification. The authors should review their initial classification of variants pathogenicity for radical changes in genes for which loss of function may not be a recognised mechanism of disease. For example, previously, MYH7 and MYL2 radical variants have not been identified to be enriched in HCM or DCM (PMID: 27532257), while their initial classification was pathogenic.

Despite of a recent article supporting the role of radical MYH7 variants in LVNC, (PMID: 33500567), the role of those variants in classic HCM remains elusive. Therefore, the classification as Pathogenic of radical variants such as, but not limited to, c.4170-2A>G and c.896-2A>G in heterozygosity each in an unrelated individuals, should be explained in relation to LVNC rather than classical HCM.

Conversely, a well known Pathogenic/Likely Pathogenic (P/LP) variant, MYH7, c.1051A>G (p.Lys351Glu) with no conflicting classification in ClinVar (https://www.ncbi.nlm.nih.gov/clinvar/variation/181335/) was classified in a VUS. Although this variant was not yet assessed by the MYH7 expert panel, for all other MYH7 variants, the authors’ should adopt the classification of the MYH7 variants present in ClinVar and reviewed by reviewed by expert panel (a FDA approved database).

Therefore, the initial classification of variants pre S:N criteria, as presented in Supply. Table 4 should be revised according to the current classification (i.e.: ClinVar) and it would be beneficial to add a column showing the new classification after applying S:N criteria.

Given the possible re-classification of some of the variants presented in Supply. Table 4, it would be interesting to know whether any subject identified with a secondary finding with HCM-associated variant, truly developed HCM or LVNC or any form of cardiomyopathy after a longer follow-up and if there was any correlation between P/LP variant type (missense vs radical), protein localisation and age of onset (inference of penetrance and severity).

Reviewer 2 Report

In this work Kurzlechner and co-workers evaluated the frequency of incidental pathogenic/likely pathogenic (P/LP) variants and variants of uncertain significance (VUS) of HCM associated genes in pediatric patients who underwent exome sequencing (ES) analysis. In order to better predict  the pathogenicity of these variants, they used a method to identify possible pathogeinc hotspot within a gene, the amino acid-level signal to noise (S:N) analysis.

MAJOR COMMENTS

Authors compared the frequency of VUS and P/LP variants in HCM patients, in ES cohort and general popilation. The gene-level S:N showed that five genes (MYH7, MYBPC3, TNNI3, TNNT2 and MYL2) had the highest probability of pathogenic desease associated variants versus physiologically tolerated population variants. Among them, MYBPC3 and MYH7 were the most significative. However, as the authors say in the discussion, it is well knonw that these genes, and in particular are MYBPC3 and MYH7, are the most commonly associated to HCM.

The amino acid level S:N analysis showed that missense variants located within 169-982 codons of MYH7 have higher pobability to be pathogenic, as well as radical variants in MYBPC3. These results have been highlighed from long time (also Richards and collegues comment the pathogenicity of these variants as examples in their guidelines) and these data are currently used for variants’ evaluation (PM1 for hot spot variants in critical domains and PVS1 for null variants).

Moreover, some results are not well shown. Supplementary table 5 is lacking; it is detalied how many variants have been re-classified but it is not clear which of them changed the classification. These details are available only for two patients of TCH cohort.

The authors also specify that some patients underwent ES for clinical features that included  cardiomiopathy, isolated or syndromic. Beyond HCM, also DCM and LVNC are phenotypes associated with LP/P variants in  some sarcomeric genes, such as MYBPC3, MYH7 and TNNI3. In my opinion, the identification of LP/P variants in patients with HCM (six, as specified in Supplementary), DCM and LVNC cannot be considered “incidental”.

What about segregations studies that authors performed? Variants were de novo in 6 patients. In the other cases they should be inherited. What about the relatives’ phenotype?

MINOR COMMENTS

Introduction. Frequency of HCM (1:500) is related to general population and in particular to adult patients. Mortality risk (bimodal peak in the first 2 years and adolescence) is referred to pediatric population. In the text these data are presented consecutively, without clear distinction. It is confounding for the reader.

Table 2. What’s the meaning of Hx? Please detail it in the legend.

Round 2

Reviewer 2 Report

The authors have improved their work by including all the proposed suggestions in the manuscript and in the supplemementary